# Extracellular Vesicle-Associated MicroRNA-138-5p Regulates Embryo Implantation and Early Pregnancy by Adjusting GPR124

**DOI:** 10.3390/pharmaceutics14061172

**Published:** 2022-05-30

**Authors:** Hsien-Ming Wu, Tzu-Chi Lo, Chia-Lung Tsai, Liang-Hsuan Chen, Hong-Yuan Huang, Hsin-Shih Wang, John Yu

**Affiliations:** 1Department of Obstetrics and Gynecology, Chang Gung Memorial Hospital Linkou Medical Center, Chang Gung University School of Medicine, Taoyuan 333, Taiwan; cltsai24@gmail.com (C.-L.T.); elsachen0119@gmail.com (L.-H.C.); hongyuan@cgmh.org.tw (H.-Y.H.); hswang0707@gmail.com (H.-S.W.); 2Institute of Stem Cell and Translational Cancer Research, Chang Gung Memorial Hospital Linkou Medical Center, Chang Gung University School of Medicine, Taoyuan 333, Taiwan; jlo0330@cgmh.org.tw (T.-C.L.); johnyu@gate.sinica.edu.tw (J.Y.)

**Keywords:** extracellular vesicle, decidua, microRNA, endometrium, GPR124, NLRP3

## Abstract

Functional embryo–maternal interactions occur during the embryo implantation and placentation. Extracellular vesicles with microRNA (miR) between cells have been considered of critical importance for embryo implantation and the programming of human pregnancy. MiR-138-5p functions as the transcriptional regulator of G protein-coupled receptor 124 (GPR124). However, the signaling pathway of miR138-5p- and GPR124-adjusted NLRP3 inflammasome activation remains unclear. In this study, we examine the roles of the miR138-5p and GPR124-regulated inflammasome in embryo implantation and early pregnancy. Human decidual stromal cells were isolated from the abortus tissue and collected by curettage from missed abortion patients and normal pregnant women at 6- to 12-week gestation, after informed consent. Isolated extracellular vesicles from decidua and decidual stromal cells were confirmed by transmission electron microscopy (TEM). Next-Generation Sequencing (NGS) and microarray were performed for miR analysis. The predicated target genes of the differentially expressed miR were analyzed to identify the target genes and their pathway. We demonstrated the down-regulation of miR-138-5p and the overexpression of GPR124 in spontaneous miscarriage compared to normal pregnancy. We also showed the excessive activation of the NLRP3 inflammasome in spontaneous miscarriage compared to normal pregnancy. Here, we newly demonstrate that the miR-138-5p and GPR124-adjusted NLRP3 inflammasome were expressed in extracellular vesicles derived from decidua and decidual stromal cells, indicating that the miR-138-5p, GPR124 and NLRP3 (NACHT, LRR, and PYD domains-containing protein 3) inflammasome have a potential modulatory role on the decidual programming and placentation of human pregnancy. Our findings represent a new concept regarding the role of extracellular vesicles, miR-138-5p, GPR124, and the NLRP3 inflammasome in normal early pregnancy and spontaneous miscarriage.

## 1. Introduction

The embryo–maternal interactions play an important role in the successful embryo implantation and pregnancy [1]. Functional embryo–maternal interactions occur during the embryo implantation and placentation [2,3]. During the process of embryo implantation, weakened rejection of the maternal immune system contributes to subsequent successful pregnancy [4]. The novel embryo/endometrial interactions through dynamic intracellular and secreted protein changes in the peri-implantation period were demonstrated to modulate the successful establishment of pregnancy [5]. The role of extracellular vesicles and microRNA (miR) in decidualization for the embryo–maternal interactions of decidual endometrial stromal cells has not been clarified in the implantation of the embryos. 

Extracellular vesicles are secreted by most cells. They usually intercede intercellular communication through the transfer of genetic information as well as the transfer of miR [6]. Extracellular vesicles with miR between cells have been considered of critical importance for embryo implantation and programming of human pregnancy [7]. In this in vitro study, we examined whether extracellular vesicles and miR could be identified from tissues and cells derived from abortus decidua. Based on the data shown in microarray analysis of miR, we demonstrate the expression of microRNA-138-5p and G protein-coupled receptor 124 (GPR124) in normal early pregnancy and spontaneous miscarriage. Next-Generation Sequencing (NGS) and microarray were performed for miRNAs analysis. Isolated extracellular vesicles from decidua and decidual stromal cells were confirmed by TEM images. The roles of some miR in specific processes associated with the pathophysiology of blood supply have been demonstrated, such as affecting angiogenesis through reducing the functions of smooth muscle cells and endothelial cells [8,9]. MiRs may be involved in inflammation, and vascular degeneration related to other mechanisms such as embryonal development or the integrity of the blood–brain barrier still needs to be elucidated. The goal of our study was to identify new miRs involved in embryo implantation and pregnancy. We demonstrated the down-regulation of miR-138-5p in spontaneous miscarriage compared to normal pregnancy. We also demonstrated the overexpression of GPR124 (target gene of miR-138a-5p) in spontaneous miscarriage compared to normal pregnancy. The importance of the NLRP3 inflammasome in immunity and human diseases has been well documented, but the signaling pathway of miR-138-5p- and GPR124-adjusted NLRP3 inflammasome activation remains unclear. In the present study, we examine the roles of miR-138-5p and GPR124-adjusted inflammasome in embryo implantation and early pregnancy. Our results imply the potential of elucidating miR-138-5p and GPR124 as a vital role in the process of embryo implantation and placentation in human.

## 2. Materials and Methods

### 2.1. Cell Culture

To clarify the roles of miR-138-5p and GPR124 in human decidual stromal cells, after informed consent was received from women aged 25–38 years undergoing surgical termination of the normal pregnancy and spontaneous miscarriage after 6–9 weeks of gestation, human decidual stromal cells were isolated from the decidual tissue. Human subjects’ involvement in this study was approved by the Institutional Review Board of Chang Gung Memorial Hospital (CGMH-IRB numbers 201601676A3, 201702112B0, 201802242A3, and 201902015B0 on 27 December 2017). Enzymatic digestion and mechanical dissociation were performed to isolate human decidual stromal cells from decidual tissue based on a modified protocol [10]. Briefly, the human decidual tissue was minced and treated with type IV collagenase (Sigma-Aldrich, St. Louis, MO, USA) and DNase type I in a shaking water bath at 37 ℃ for 90 min. The cell digest was then passed through a 70 μm filter. The decidual epithelial and stromal cells were collected in a 50 mL tube. Next, epithelial cells were separated from stromal cells with a 45 μm filter. The stromal cells were subsequently pelleted by centrifugation at 1000× *g* for 5 min at room temperature. The cell pellet was washed once, resuspended and plated in Dulbecco’s Modified Eagle Medium (DMEM) containing 25 mM glucose, L-glutamine, and antibiotics (100 U/mL penicillin and 100 μg/mL streptomycin) and supplemented with 10% fetal bovine serum. 

### 2.2. Extracellular Vesicle Isolation 

For the collection of conditioned medium, the primary endometrial stromal cells were grown in DMEM containing 25 mM glucose, 200 mM *L*-glutamine, and antibiotics (100 U/mL penicillin and 100 μg/mL streptomycin), and they were supplemented with 10% fetal bovine serum (FBS) (Thermo Fisher, Waltham, MA, USA) at 37 °C, 5% CO_2_, following which the medium was changed to Gibco™ Extracellular vesicle-Depleted FBS (Thermo Fisher, Waltham, MA, USA, Catalog # A2720801) for 72 h, and the supernatants were collected for extracellular vesicles isolation. Briefly, the culture medium was centrifuged at 1200× *g* for 10 min to remove contaminating cells, which was followed by filtration through 0.22 μm filters to remove cell debris and particles larger than 200 nm. Then, it was centrifuged twice at 10,000× *g* for 30 min to remove larger microvesicles (MVs). Extracellular vesicles were pelleted by ultracentrifugation at 100,000× *g* for 60 min (Optima XE-90, Beckman Coulter, rotor Ti SW28, Kraemer Blvd., Brea, CA, USA) (all steps were performed at 4 °C). The extracellular vesicles were dissolved in 100 μL of Phosphate buffered saline (PBS) (Thermo Fisher, Waltham, MA, USA) solution to prepare a suspension, which was stored at −80 °C. 

### 2.3. Extracellular Vesicle RNA Isolation

TRI Reagent LS (TRIzol Reagent liquid samples) (750 μL, Sigma, Catalog Number T3934) was added to 250 µL of extracellular vesicle sample. Subsequently, 200 µL of chloroform (Thermo Fisher, Waltham, MA, USA) was added to each sample and mixed thoroughly by shaking for over 30 s and incubated at room temperature for 10 min. Phase separation was performed by centrifugation at 13,000 rpm at 4 °C for 15 min. The upper aqueous phase was collected. One microliter of glycogen (20 μg/μL) (Sigma Aldrich) and equal isopropanol for ribonucleic acid (RNA) precipitation were added to each sample. Samples were mixed, incubated 1.5 h at −80 °C and then centrifuged at 13,000 rpm at 4 °C for 30 min to pellet RNA. The pellet was washed with 75% ethanol and centrifuged at 13,000 rpm at 4 °C for 10 min. Pellets were dried for 10 min before being resuspended in 15 μL of Diethyl pyrocarbonate (DEPC) water, and purified RNA was quantified using Nanodrop (Thermo Fisher, Waltham, MA, USA). Total extracellular vesicle RNAs were reverse transcribed to complementary deoxyribonucleic acid (cDNA) using a TaqMan™ MicroRNA Reverse Transcription Kit according to the manufacturer’s instructions. Quantitative polymerase chain reaction (PCR) was performed using TaqMan™ MicroRNA assay (Applied Biosystems, Waltham, MA, USA) in Applied Biosystems™ QuantStudio™ 5 Real-Time PCR System (Thermo Fisher, Waltham, MA, USA). All of the TaqMan™ MicroRNA assay (hsa-miR-138-5P: 0022084, U6 snRNA: 001973) was purchased from Applied Biosystems. U6 snRNA served as internal controls for miR. Quantitative PCR conditions were: pre-denaturation at 95 °C, 10 min followed by 95 °C 15 s, 60 °C 60 s, and a total of 40 cycles to detect the expression of miR-138-5P and U6 snRNA. Expression levels are triplicated as the cycle threshold (Ct) value of the candidate gene relative to the Ct value of the housekeeping gene. Fold changes in the expression of each miR were calculated by a comparative threshold cycle (Ct) method using the formula: ΔCt = Ct_miRNA_ − Ct_U6_, ΔΔCt = ΔCt_case extracellular vesicle-miRNA_ − ΔCt_normal human exosoem-miRNA_. (ΔCt: the difference in Ct values for the target miRNA and internal control; ΔΔCt: the comparative CT).

### 2.4. Nanoparticle Tracking Analysis (NTA)

Isolated extracellular vesicles were analyzed using Nanosight NS300 (Malvern Instruments, Worcestershire, UK) equipped with a blue laser (488 nm). Nanoparticles were illuminated by the laser, their movement under Brownian motion was captured for 60 s, and the video recorded was subjected to NTA using the Nanosight particle tracking software to calculate nanoparticle concentrations and size distribution. The concentration of extracellular vesicles through NTA is 3.93 × 10^12^ ± 2.19 × 10^11^ particles/mL.

### 2.5. Extracellular Vesicle Visualization by (TEM) and Immunogold-EM

The sample was fixed with 2.5% glutaraldehyde (Sigma Aldrich) for 30 min at room temperature. Then, 10–20 μL of extracellular vesicle suspension was absorbed onto formvar carbon-coated copper electron microscopy grids (300 mesh) at room temperature for 5 min, and it was then subjected to 3% uranyl acetate (Sigma Aldrich) staining for 30 min. Grids were washed three times with PBS and were maintained in a semi-dry state before observation by TEM (JEOL JEM-1400, Tokyo, Japan). Extracellular vesicle features were analyzed by the immunogold-EM as described previously [11,12]. In brief, for the detection of cluster of differentiation 63 and 81 (CD63 and CD81)of extracellular vesicles by immunogold-EM, the blocked grids, containing fixed exosomes with 2% paraformaldehyde (PFA) (Sigma Aldrich), were transferred to the drops of the anti-CD63 and anti-CD81 antibody (dilution = 1:100) in PBS/0.5% bovine serum albumin (BSA), and they were further incubated for 1 h. The extracellular vesicles on the grids were then rinsed with PBS/0.5% BSA five times for 3 min, which was followed by their incubation with 10 nm gold-labeled secondary antibody in PBS/0.5% BSA for 30 min, and then, they were rinsed five times for 3 min with 100 µL drops of PBS/0.5% BSA. The extracellular vesicles on the grids were stained with 2% uranyl acetate and then inspected under a TEM.

### 2.6. miRNA Microarray

The total RNA of endometrial stromal cells was purified by Quick-RNA^TM^ MicroPrep Kit (Zymo Research, Irvine, CA, USA). The miRNA expression profiles were analyzed using the Affymetrix Gene-Chip miRNA 4.0 array. Total RNA (1 μg) including miRNA from tissue was biotin-labeled using the FlashTag^TM^ Biotin HSR (Thermo Fisher, Waltham, MA, USA) RNA Labeling. The samples were hybridized using GeneChip^®^ Hybridization Oven (Thermo Fisher, Waltham, MA, USA) to the Affymetrix miRNA microarray according to the protocols provided by the manufacturer. The labeled RNA was heated to 99 °C for 5 min and then to 45 °C for 5 min. RNA-array hybridization was performed with agitation at 60 rotations per minute for 16 h at 48 °C on an Affymetrix^®^ 450 Fluidics Station (Thermo Fisher, Waltham, MA, USA). The chips were washed and stained using a Genechip Fluidics Station. The hybridized chips were scanned by a GeneChip Scanner 3000 7G (Thermo Fisher, Waltham, MA, USA). Signal values were computed using the Affymetrix^®^ GeneChip™ Command Console software 3.0. (Thermo Fisher, Waltham, MA, USA).

### 2.7. Reverse Transcription-Quantitative Polymerase Chain Reaction

Total extracellular vesicle RNAs were reverse transcribed to cDNA using a TaqMan™ MicroRNA Reverse Transcription Kit (Applied Biosystems PN: 4366596) with primers for miR-138-5p and U6 small nuclear RNA (hsa-miR-138-5p: assay ID: 002284, U6 snRNA: assay ID:001973, Applied Biosystems) according to the manufacturer’s instructions. Quantitative PCR was performed using a QuantStudio™ 5 Real-Time PCR System (Thermo Fisher, Waltham, MA, USA). Real-time PCR reactions with cDNAs were performed in a 20 μL final volume. A reaction mix containing 2 μL of cDNAs, 10 μL 2× TaqMan^®^ Universal PCR MasterMix II, no UNG (#4440040) and 1 μL of 1×TaqMan™ MicroRNA assay (20×; #4427975) and 7 μL DEPC-water was loaded in each well. All of the TaqMan™ MicroRNA assays (Applied Biosystems) (hsa-miR-138-5p: assay ID: 002284, U6 snRNA: assay ID:001973, Applied Biosystems) were purchased from Life Technologies (Carlsbad, CA, USA). U6 snRNA served as internal controls for miRNA. Quantitative PCR conditions were: pre-denaturation at 95 °C, 10 min followed by 40 cycles of 95 °C for 15 s and 60 °C for 1 min to detect the expression of miR-138-5P and U6 snRNA. Expression levels are triplicated as the cycle threshold (Ct) value of the candidate gene relative to the Ct value of the housekeeping gene. Fold changes in the expression of each miRNA were calculated by a comparative threshold cycle (Ct) method using the formula: ΔCt = Ct_miRNA_ − Ct_U6_, ΔΔCt = ΔCt_case exosome-miRNA_ − ΔCt_normal human exosoem-miRNA_. A list of the TaqMan^®^ miRNAs and U6 snRNAs used in this study is provided. 

### 2.8. QIASeq miRNA Data Analysis and Target Gene Prediction

The analysis of Unique Molecular Index (UMI) counts to calculate changes in miRNA expression was performed via Primary QIAseq miRNA Quantification Data Analysis software followed by Secondary QIASeq miRNA Library Kit Data Analysis Software (Qiagen, Hilden, Germany). Fold regulation and fold change in the miRNA dataset were obtained upon the normalization by the geNorm method. It has been determined that the results of cross multiple prediction algorithms can increase specificity and decrease sensitivity. Hence, we chose to integrate the results of the bioinformatics prediction programs TargetScan 7.2 (http://www.targetscan.org accessed on 2 March 2018) and MiRDB (http://mirdb.org/miRDB/ accessed on 2 March 2018). Taken together, GPR124 was eventually determined by two programs to be the target of miR-138-5p.

### 2.9. Immunoblot Analysis 

The cells were lysed in buffer containing 20 mM Tris, pH 7.4, 2 mM Ethylene glycol-bis(2-aminoethylether)-*N*,*N*,*N*′,*N*′-tetraacetic acid (EGTA), 2 mM Na_2_VO_3_, 2 mM Na_4_P_2_O_7_, 2% Triton X-100, 2% Sodium dodecyl sulfate (SDS), 1 μM aprotinin, 1 μM leupeptin and 1 mM phenylmethanesulfonylfluoride or phenylmethylsulfonyl fluoride (PMSF). The protein concentration was determined with a protein assay kit using BSA standards according to the manufacturer’s instructions (Bio-Rad Laboratories, Hercules, CA, USA). Equal amounts of cell lysate were separated by SDS polyacrylamide gel electrophoresis (PAGE) and transferred to a nitrocellulose membrane (Hybond-C, Amersham Pharmacia Biotech Inc., Oakville, ON, USA). Following blocking with Tris-buffered saline (TBS) containing 5% non-fat dry milk for 1 h, the membranes were incubated overnight at 4 °C with anti-TSG101 (Cell Signaling, Danvers, MA, USA), anti-CD63 (Cell Signaling), anti-CD9 (Cell Signaling), anti-CD81 (Cell Signaling), anti-GPR124 (Cell Signaling), anti-NLRP3 (Cell Signaling), anti-IL-18 (Cell Signaling), or anti-IL-1β (Cell Signaling) antibody followed by incubation with HRP-conjugated secondary antibody. The immunoreactive bands were detected with an enhanced chemiluminescence (ECL) kit. The membrane was then stripped with stripping buffer (62.5 mM Tris, 10 mM DTT, and 2% SDS, pH 6.7) at 50 °C for 30 min and re-probed with anti-β-actin, anti-GAPDH, and anti-α-tubulin antibody (Santa Cruz, Dallas, TX, USA) as a loading control.

### 2.10. Immunohistochemistry (IHC)

To demonstrate the expression of the GPR124 protein in human decidual endometrial tissue, we performed IHC on sections of human decidual endometrial tissue using previously reported procedures [13]. The involvement of human subjects in this study was approved by the Institutional Review Board of Chang Gung Memorial Hospital (CGMH-IRB numbers: 201601676A3, 201702112B0, 201802242A3, and 201902015B0 27 December 2017). Four-micrometer-thick formalin-fixed, paraffin-embedded (FFPE) tissue sections were deparaffinized in xylene and rehydrated with a graded series of ethanol solutions. The sections were then stained with an anti-human GPR124 polyclonal antibody (Neomarker; 1:100) (Thermo Fisher, Waltham, MA, USA) using an automated IHC stainer with the Ventana Basic DAB Detection kit (Tucson, AZ, USA) according to the manufacturer’s protocol. Counterstaining was performed with hematoxylin. Sections were stained without the GPR124 antibody as a negative control in the third of three columns depicting the human decidual endometrial tissue sections.

### 2.11. Small Interfering RNA Transfection 

siGENOME ON-TARGETplus SMARTpool human GPR124 siRNA and siCONTROL NON-TARGETINGpool siRNA were purchased from Dharmacon (Lafayette, CO, USA). The cells were transfected with siRNA (50 nM) using Lipofectamine RNAiMAX. After a 48 h transfection, the medium was removed and changed to fresh serum-free medium. To examine the siRNA transfection, cells were transfected with 50 nM si-GLO (Dharmacon) for 48 h. The transfection efficiency was examined by fluorescent microscopy.

### 2.12. Enzyme-Linked Immunosorbent Assay (ELISA)

Human primary endometrial stromal cells were transfected with miR-138-5P (25 nM) and si-GPR124 (50 nM) for 48 h. The endometrial stromal cell medium was collected at 48 h after transfection to assess Interleukin (IL-1β, IL-18), and NLRP3 levels. The protein levels of human IL-1β (EZ-Set^TM^ ELISA Kit:EZ0392) (St. Louis, MO, USA) and IL-18 (PicoKine^TM^ ELISA:EK0864) (St. Louis, MO, USA) were measured in the endometrial stromal cell supernatants using ELISA kits from BOSTER ELISA kit. Human NLRP3 (ab274401) were detected using abcam ELISA kits (St. Louis, MO, USA). ELISAs were performed following the manufacturer’s instructions.

### 2.13. Protein Quantification Assays 

Cell culture supernatants were surveyed, at the demonstrated times, for the existence of inflammasome activity by using mouse GPR124, NLRP3, IL-18, and IL-1β DuoSet ELISA (R&D and MBL) (St. Louis, MO, USA) according to the manufacturer’s instructions. adenosine triphosphate (ATP) was quantified in cell supernatants using ATP Determination Kit (Life Technologies).

### 2.14. Statistical Analysis

The results are shown as the means ± SEM. Statistical evaluation was conducted with the *t*-test for paired data. Multiple comparisons were first analyzed by one-way ANOVA, followed by Tukey’s multiple comparison tests. A significant difference was defined as *p* < 0.05.

## 3. Results

### 3.1. Extracellular Vesicle Isolation and Characterization

By using nanoparticle tracking analysis (NTA), a created and verified approach [14,15], the features of separated and purified extracellular vesicles from endometrial stromal cells (ESC) are demonstrated in Figure 1A. NTA recognized particles with a diameter of 40 to 130 nm (Figure 1A). TEM recognized vesicles with a round shape appearance, which is the feature of extracellular vesicles (Figure 1B). Furthermore, extracellular vesicles were shown with positive extracellular vesicle protein markers CD63, TSG101, CD9, and CD81 presented in Figure 1C [16]. CD63 and CD81 positive dots of extracellular vesicles were analyzed by TEM and immunogold-EM in Figure 1D.

### 3.2. Microarray Analysis of miRNAs for Endometrial Tissue

The total RNA of endometrial stromal cells was purified. The miRNA expression profiles were analyzed using the Affymetrix Gene-Chip miRNA 4.0 array (Thermo Fisher, Waltham, MA, USA). Total RNA (1 μg) including miRNA from tissue was biotin-labeled using the FlashTag^TM^ Biotin homogeneously staining regions (HSR) RNA Labeling. In Figure 2A, the differential expression of miRNAs between normal pregnancy and spontaneous miscarriage in a hierarchical clustering analysis is demonstrated. The red color indicates up-regulation, and the green color indicates down-regulation. In Figure 2B, a volcano plot of microarray data shows the differences between normal pregnancy and spontaneous miscarriage. Differentially expressed miRNAs with a fold change ≥1.5 and *p* < 0.05 between the two groups are shown in red and green. In Figure 2C, Quantitative reverse transcription PCR (RT-qPCR) analysis of the expressed miRNA showed that miR-138-5p is down-regulated in spontaneous miscarriage compared to normal pregnancy.

### 3.3. To Identify That GPR124 Is a Direct Target of miR-138-5p 

To identify miRNA expression signatures, we performed miRNA microarray analysis in endometrial tissue. Hierarchical clustering analysis revealed a significant down-regulation of eight miRNAs and up-regulation of seven miRNAs in spontaneous miscarriage ESC compared to normal pregnancy ESC in Figure 2A. Gene target identification for miRNAs was conducted using the Target scan. Target scan, a miRNA target prediction program, is applied to search for putative target genes of miR-138-5p. By using Target scan, GPR124 was predicted as a potential target of miR-138-5p in Figure 3A. Target Scan analysis forecasted one putative miR-138-5p binding site within the 3′ UTR of GPR124 in Figure 3B. We focused on GPR124 because of its established relevance in angiogenesis and atherosclerosis possibly related to embryo implantation. 

### 3.4. GRP124 RNA Expressions in Endometrial Stromal Cells

To evaluate GRP124 RNA expressions in endometrial stromal cells, the immunohistochemistry of decidual endometrium sections was performed in Figure 4A. In column 1, hematoxylin and eosin staining is shown. GRP124 (brown staining) stromal cells (arrow) are indicated in rows 2 and 3 of column 2. Second antibody-only control was shown in column 3. Micrographs were taken with the 400× objective lens. Scale bars represent 20 μm. The expressions of the GRP124 protein in endometrial stromal cells are significant in spontaneous miscarriage compared to normal pregnancy. In Figure 4B, by using qRT-PCR and immunoblot analysis, the expressions of GRP124 mRNA and protein in endometrial stromal cells of normal early pregnancy and spontaneous miscarriage were detected. In Figure 4C, in the left column, the expression of GPR124 regulated by miR-138-5p at the mRNA level is shown. In the right column, the expression of GPR124 regulated by miR-138-5p at protein level is shown. Analysis for the expression of miR-138-5p and GPR124 mRNA and protein showed a significant and inverse correlation between miR-138-5p and GPR124 in endometrial stromal cells.

### 3.5. The Expression of NLRP3, IL-18, and IL-1β Inflammasome in Endometrial Stromal Cells

To analyze GRP124 and inflammasome expressions regulated by miR138-5p and GRP124 in endometrial stromal cells isolated from the decidual tissue of early pregnancy undergoing surgical termination, the expressions of GRP124 and inflammasome NLRP3, IL-18 and IL-1β were evaluated by ELISA. In Figure 5A, by using ELISA, the expressions of GRP124 and NLRP3 regulated by miR-138-5p in endometrial stromal cells were demonstrated. The expressions of IL-18 and IL-1β controlled by miR-138-5p in endometrial stromal cells are shown in Figure 5B. In Figure 5C, by using ELISA, the expressions of GRP124 and NLRP3 regulated by si-GPR124 in endometrial stromal cells are shown. The expressions of IL-18 and IL-1β regulated by si-GPR124 in endometrial stromal cells are shown in Figure 5D. By using immunoblotting, the expressions of GRP124, NLRP3, IL-18 and IL-1β protein were abolished after pretreatment of the endometrial stromal cells with miR-138-5p in Figure 6A. In Figure 6B, after pretreatment of the endometrial stromal cells with GPR124 siRNA, the protein expressions of GRP124, NLRP3, IL-18 and IL-1β were significantly reduced.

## 4. Discussion

The embryo–maternal interactions during embryo implantation and early pregnancy are important in the regulation of human reproduction. Extracellular vesicles with (miRNA) between cells have been considered of critical importance for embryo implantation and the programming of human pregnancy. The aim of this study was to investigate the functional embryo–maternal interactions occurring during the embryo implantation and placentation through extracellular vesicles with miRNA. Extracellular vesicles dedicate to the complex biological processes of cell–cell interactions. The present evidence shows that extracellular vesicles merge into the plasma membrane of the target cell and release their contents such as miRNAs into the cell, where they could immediately interact with the target cell [17,18]. Our results demonstrated that extracted extracellular vesicles from decidua and decidual stromal cells were examined by nanoparticle tracking analysis, TEM, and immunoblot analysis. These extracellular vesicles may contain some miRNAs that can contribute to embryo implantation and pregnancies. MiRNAs are groups of non-coding RNA that function to regulate the expression of target genes by fully or partly binding to their individual complementary sequences [19,20]. The expression profiles of miRNAs in the process of embryo implantation and early pregnancy may reveal some pathological modification, possibly specific to the spontaneous miscarriage. The actions of some miRNAs in specific processes concerning to the pathophysiology of spontaneous miscarriage have been demonstrated following some studies, such as affecting angiogenesis by increasing apoptosis and reducing the proliferation and migration of endothelial cells and smooth muscle cells [8,9,20]. The participation of miRNAs in the adjustment of the underlying signaling pathway related to abnormal embryo implantation and pregnancy still needs to be elucidated. We demonstrated the differential expression of miRNAs between normal pregnancy and spontaneous miscarriage in a hierarchical clustering analysis by microarray analysis of miRNAs in endometrial tissue. We showed that miR-138-5p is down-regulated in spontaneous miscarriage compared to normal pregnancy. MiR-138-5p not only has an important role in the brain but also in peripheral tissues [21]. For example, miRNA-138-5p avoided hypoxia-prompted apoptosis through the Mixed Lineage Kinase–c-Jun N-Terminal Kinase (MLK3/JNK/c-jun) pathway, subsequently resulting in protection to cardiomyocytes [22]. Furthermore, Li et al. demonstrated that miRNA-138-5p participated in the suppression of apoptosis of pulmonary artery smooth muscle cell via the disruption of caspases activation and B-cell lymphoma 2 (Bcl-2) signaling by targeting Macrophage Stimulating 1 (Mst1) [23]. Moreover, gene ontology analysis showed that miR-138-5p modulates physiological actions such as angiogenesis, cell migration, and placenta establishment, indicating that the dysregulation of these physiological actions has been related to abnormal embryo implantation and pregnancy [24,25]. To explore target genes of miR-138-5p, we searched for putative targets using established miRNA target prediction programs, Target scan. GPR124 was predicted as a potential target of miR-138-5p by Target scan. GPR124 functionally promotes physiological embryonic and angiogenesis within the central nervous system [26,27]. We focused on GPR124 because of its established relevance in angiogenesis and atherosclerosis possibly related to embryo implantation. Experimental validation exhibited that the expression of GPR124 was inhibited by miR-138-5p on protein and mRNA levels in cancer cells [28]. Our results showed an inverse relation between the expression of miR-138-5p and GPR124 in endometrial tissues. Knockdown of GPR124 mimicked the effects of miR-138-5p on the angiogenesis and cell migration. Taken together, our results propose that the down-regulation of miR-138-5p and activation of GPR124 participate in spontaneous miscarriage, and that the restoration of miR-138-5p and suppression of GPR124 might work for potential therapeutic approach for overcoming spontaneous miscarriage.

Inflammasomes with cytoplasmic supramolecular complexes play an important role in host immune response against external stresses such as bacterial infections [29,30,31]. They are also related to the pathogenesis of dysregulated inflammatory response, including atherosclerosis and Alzheimer diseases [32,33]. Inflammasomes initiate inflammatory caspases, which subsequently activate proinflammatory cytokines interleukin 18 (IL-18) and IL-1β and mediate the cytokine release and programmed necrotic cell death [34,35]. We demonstrated the excessive activation of inflammasome NLRP3, IL-18, and IL-1β in spontaneous miscarriage compared to normal pregnancy. The expressions of NLRP3, IL-18 and IL-1β are regulated by miR-138-5p and GRP124 in endometrial stromal cells. The excessive activation of the NLRP3 inflammasome is directly linked to systematic inflammation [36], suggesting that inflammasome NLRP3, IL-18, and IL-1β could be considered as a possible factor in spontaneous miscarriage. However, the potential signaling pathway intervened by miR-138-5p and GRP124 to control the inflammasome NLRP3, IL-18, and IL-1β in human decidual endometrial stromal cells related to embryo implantation and pregnancy is not well developed. In the present study, we demonstrate for the first time that miR-138-5p and GRP124 may be involved in early pregnancy by regulating the expression of inflammasome NLRP3, IL-18, and IL-1β, raising a realization of the process of embryo implantation and early pregnancy.

In this study, the treatment of human decidual endometrial stromal cells with miR138-5p resulted in significantly decreased expression of GPR124, NLRP3, IL-18, and IL-1β. These findings suggest that mi138-5p directly regulates the expression of GPR124, NLRP3, IL-18, and IL-1β in human decidual endometrial stromal cells. In the present study, GRP124 siRNA was used to selectively knock down the protein expression of GRP124 siRNA in human decidual endometrial stromal cells. In contrast, GRP124 with siRNA invalidated the GRP124-regulated expression of NLRP3, IL-18, and IL-1β in human decidual endometrial stromal cells, indicating that the effects of GRP124 on spontaneous miscarriage are through the expression of inflammasome NLRP3, IL-18, and IL-1β. In aggregate, attempts to manipulate the GPR124 and inflammasome NLRP3, IL-18, and IL-1β signaling that have a critical role in spontaneous miscarriage may be a way to clarify the actions of mi138-5p in decidual endometrial stromal cells related to embryo implantation and early pregnancy. This is one of the novel findings in the present study. Through investigating both the bio- and physiochemical characteristics of extracellular vesicles at the microscale, the dramatic advances in microfabrication methods have provided the dawn to exploit lab-on-a-chip-type microfluidic systems for productive extracellular vesicle purification [37,38]. Microfluidic techniques are effectively providing a new insight of extracellular vesicle-associated examination by changing the extracellular vesicle isolation and characterization to an integrated one-step method [38]. This is a potential therapeutic intervention and pharmaceutics application for the treatment of human disease [39,40,41].

## 5. Conclusions

Our results demonstrate an important function for mi138-5p in regulating the embryo implantation and early pregnancy through the modulation of GPR124, and the following activation of inflammasome NLRP3, IL-18, and IL-1β signaling in human decidual stromal cells from the normal pregnancy and abnormal pregnancy (Figure 7). These data imply that the mi138-5p regulating GPR124 and inflammasome NLRP3, IL-18, and IL-1β in the decidual endometrium is a possible therapeutic target for improving the rate of embryo implantation in the treatment of infertility, and they provide a new concept regarding the signaling pathway of embryo implantation and decidual programming of human pregnancy.

## Figures and Tables

**Figure 1 pharmaceutics-14-01172-f001:**
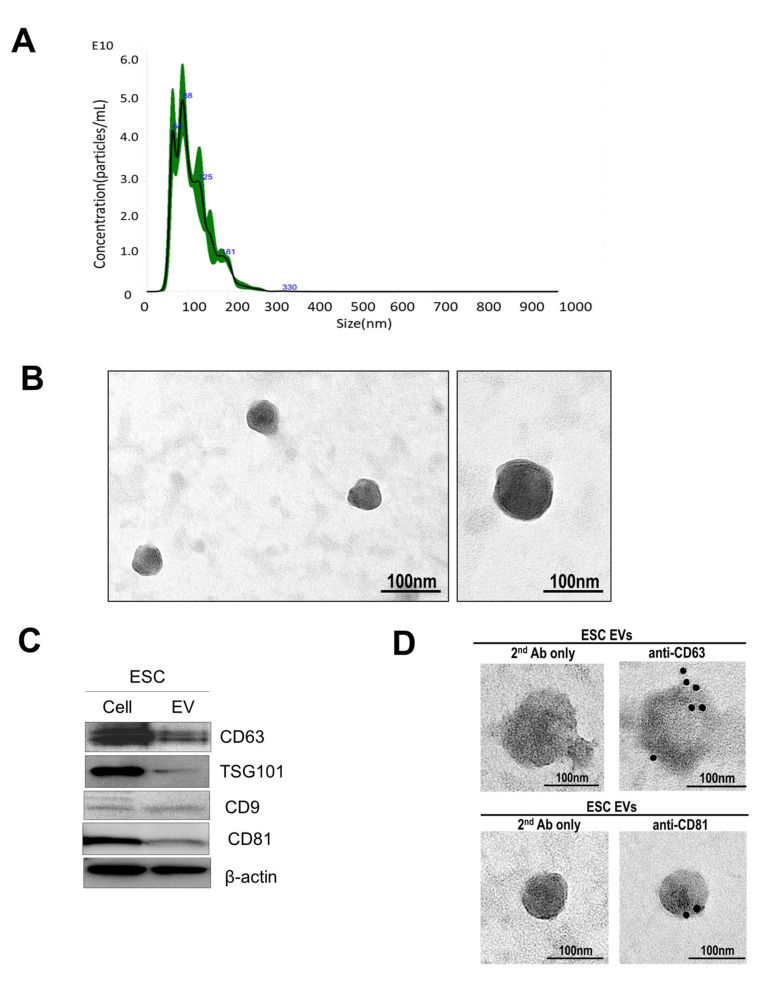
To examine the extracellular vesicles that exist in endometrial stromal cells by NTA (nanoparticle tracking analysis), TEM and immunoblot analysis. (**A**) Nanoparticles were illuminated by the laser; their movement under Brownian motion was captured for 60 s, and the video recorded was subjected to NTA using the Nanosight particle tracking software to calculate nanoparticle concentrations and size distribution. Blue numbers mean the size(nm) of extracellular vesicles. (**B**) The isolation of extracellular vesicles derived from endometrial stromal cells and villi tissues by ultra-centrifuge. In (TEM) observation, the size of ESC-extracellular vesicles was 50–150 nm. (**C**) The extracellular vesicles supernatant was denatured in 4x sodium dodecyl sulfonate (SDS) buffer and subjected to Western blot analysis using rabbit antibody CD63, CD9, CD81 and TSG101. beta-actin served as a control. (**D**) CD63 and CD81 positive dots of extracellular vesicles were analyzed by TEM and immunogold-EM.

**Figure 2 pharmaceutics-14-01172-f002:**
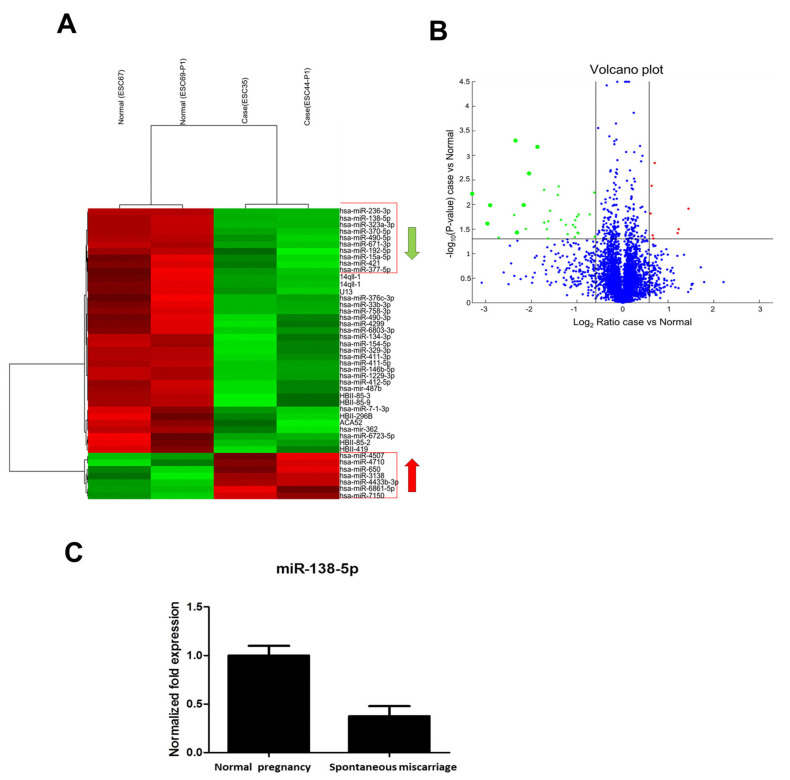
The microarray analysis of miRNAs for endometrial tissue. Total RNA of endometrial stromal cells was purified. The miRNA expression profiles were analyzed using the Affymetrix Gene-Chip miRNA 4.0 array. (**A**) Differential expression of miRNAs between normal pregnancy and spontaneous miscarriage in a hierarchical clustering analysis. The red color indicates up-regulation, and the green color indicates down-regulation. To identify miRNA expression signatures, we performed miRNA microarray analysis in endometrial tissue. Hierarchical clustering analysis revealed a significant down-regulation of 8 miRNAs and up-regulation of 7 miRNAs in spontaneous miscarriage ESC compared to normal pregnancy ESC. (**B**) Volcano plot of microarray data. Plot shows differences between normal pregnancy and spontaneous miscarriage. Differentially expressed miRNAs with a fold change ≥1.5 and *p* < 0.05 between the two groups are shown in red and green. (**C**) RT-qPCR analysis of the expressed miRNA. Total extracellular vesicle RNAs were reverse transcribed to cDNA with primers for miR-138-5p and U6 small nuclear RNA. miR-138-5p is down-regulated in spontaneous miscarriage compared to normal pregnancy.

**Figure 3 pharmaceutics-14-01172-f003:**
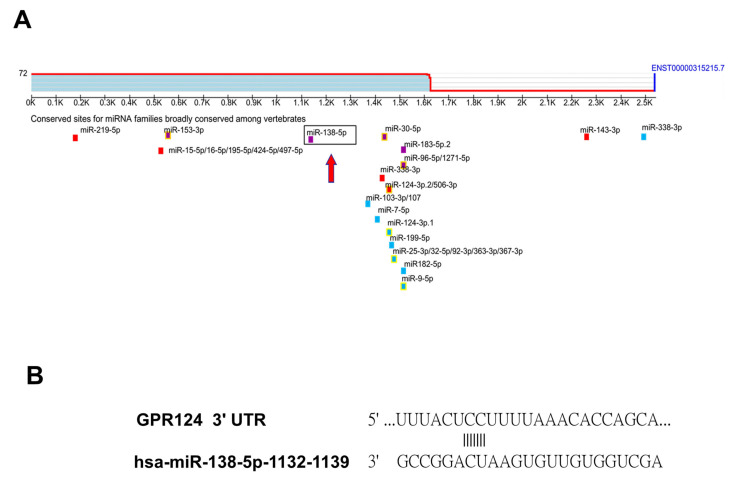
To identify that GPR124 is a direct target of miR-138-5p by Microarray and Target Scan. (**A**) After the identification of miRNA expression signatures by miRNA microarray analysis in endometrial tissue, we searched for putative targets using established miRNA target prediction programs, target Scan 7.2, a web server, in order to explore target genes of miR-138-5p. GPR124 was predicted as a potential target of miR-138-5p by target Scan. We focused on GPR124 because of its established relevance in angiogenesis and atherosclerosis possibly related to embryo implantation. (**B**) Target Scan analysis forecasted one putative miR-138-5p binding site within the 3′ UTR of GPR124.

**Figure 4 pharmaceutics-14-01172-f004:**
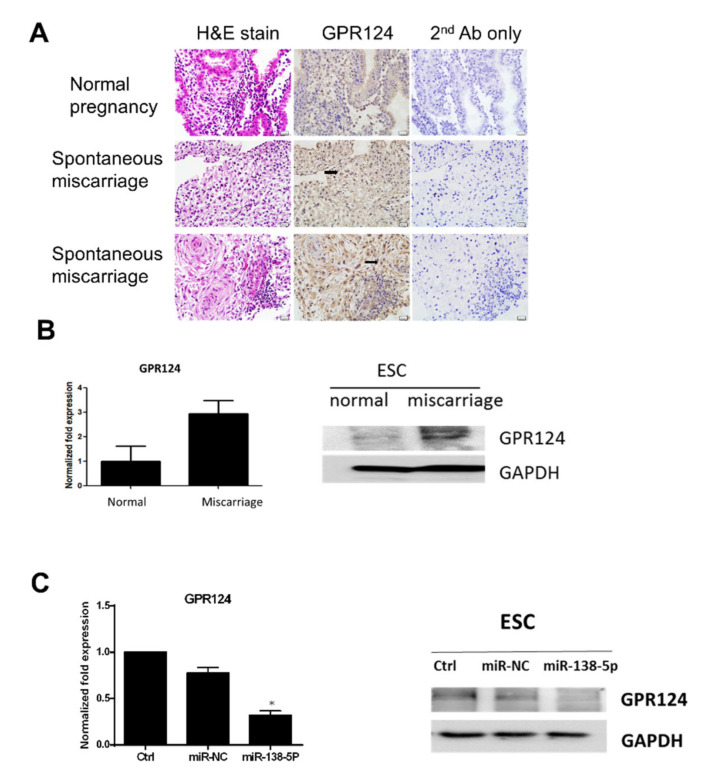
GPR124 expressions in endometrial stromal cells are examined by immunohistochemistry, quantitative real-time PCR and immunoblot analysis. (**A**) Immunohistochemistry of decidual endometrium sections; column 1, hematoxylin and eosin staining; column 2, GPR124 (brown staining) stromal cells (arrow) are indicated in row 2; column 3, second antibody only control. Micrographs were taken with the 400× objective lens. Scale bars represent 20 μm. (**B**) The expressions of GPR124 mRNA and protein in endometrial stromal cells of normal early pregnancy and spontaneous miscarriage were detected by RT-qPCR and immunoblot analysis. (**C**) In the left column, the expression of GPR124 regulated by miR-138-5p at the mRNA level. In the right column, the expression of GPR124 regulated by miR-138-5p at the protein level. Analysis for expression of miR-138-5p and GPR124 mRNA and protein showed a significant and inverse correlation between miR-138-5p and GPR124 in endometrial stromal cells. miRNA negative control (miR-NC). ESC—endometrial stromal cells. H&E—hematoxylin and eosin. Ctrl—control. Ab—antibody. (* *p* < 0.05, versus control).

**Figure 5 pharmaceutics-14-01172-f005:**
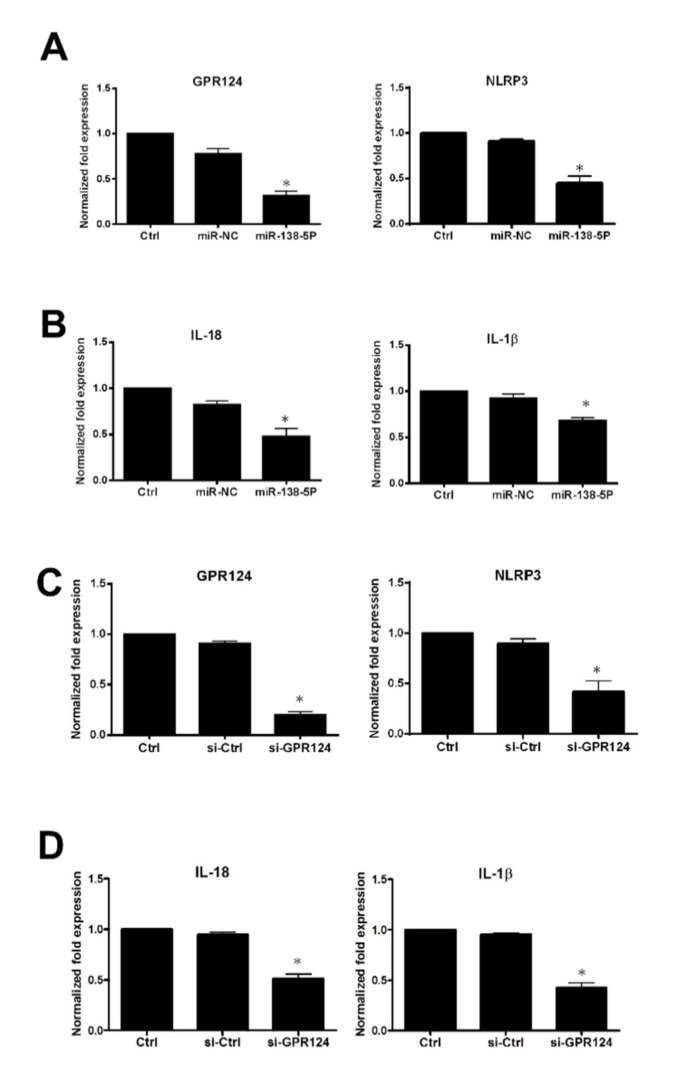
The excessive activation of NLRP3, IL-18, and IL-1β inflammasome in spontaneous miscarriage compared to normal pregnancy. Human primary endometrial stromal cells were transfected with miR-138-5P (25 nM) and si-GPR124 (50 nM) for 48 h. The endometrial stromal cell medium was collected at 48 h after transfection to assess IL-1β, IL-18, and NLRP3 levels. (**A**) The expressions of GPR124 and NLRP3 mRNA regulated by miR-138-5p in endometrial stromal cells. (**B**) The expressions of IL-18 and IL-1β mRNA regulated by miR-138-5p in endometrial stromal cells. (**C**) The expressions of GPR124 and NLRP3 mRNA regulated by si-GPR124 in endometrial stromal cells. (**D**) The expressions of IL-18 and IL-1β mRNA regulated by si-GPR124 in endometrial stromal cells. miRNA negative control (miR-NC). siRNA control (si-Ctrl). (* *p* < 0.05, versus control).

**Figure 6 pharmaceutics-14-01172-f006:**
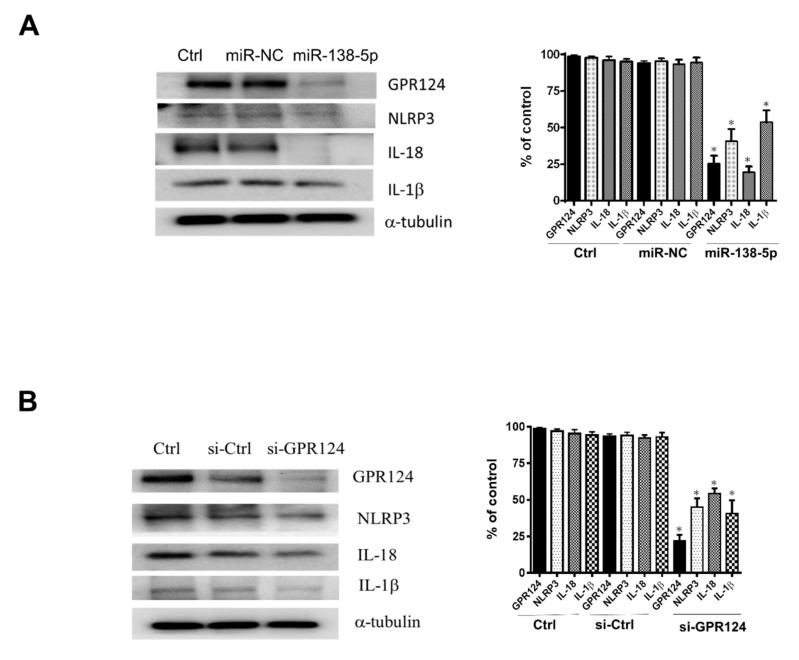
The protein expression of NLRP3, IL-18, and IL-1β inflammasome in spontaneous miscarriage compared to normal pregnancy. Human primary endometrial stromal cells were transfected with miR-138-5P (25 nM) and si-GPR124 (50 nM) for 48 h. Immunoblot analysis was performed to examine the protein levels of GPR124, IL-1β, IL-18, and NLRP3. (**A**) The expressions of GPR124, NLRP3, IL-18 and IL-1β protein regulated by miR-138-5p in endometrial stromal cells. (**B**) The expressions of GPR124, NLRP3, IL-18 and IL-1β protein regulated by si-GPR124 in endometrial stromal cells. miRNA negative control (miR-NC). siRNA control (si-Ctrl). (* *p* < 0.05, versus control).

**Figure 7 pharmaceutics-14-01172-f007:**
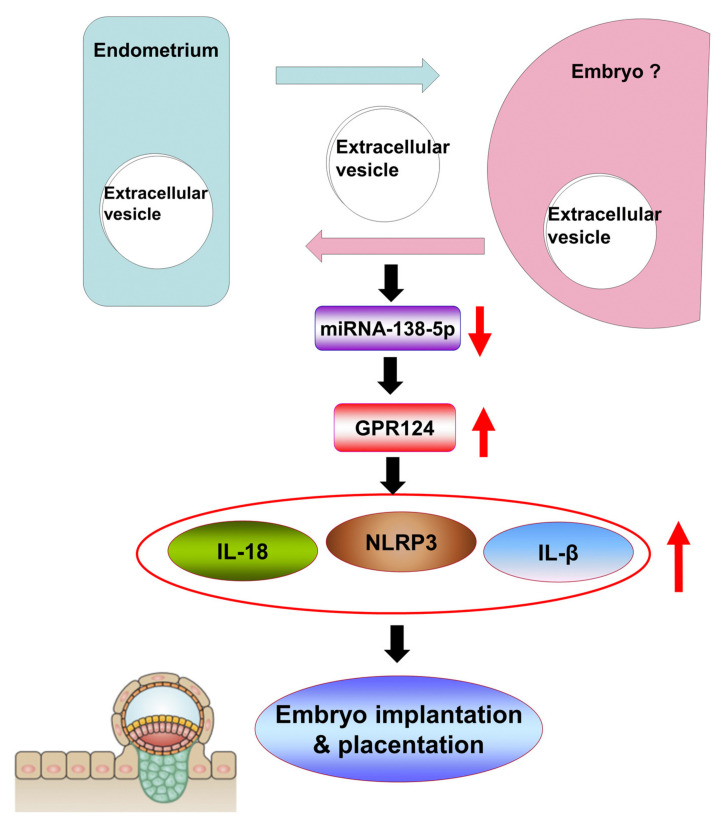
The proposed signaling pathways involved in the role of extracellular vesicles, miR-138-5p, GPR124, and NLRP3 inflammasome in normal early pregnancy and spontaneous miscarriage. These results demonstrate an important function for mi138-5p in regulating the embryo implantation and early pregnancy, through the modulation of GPR124, and the following activation of inflammasome NLRP3, IL-18, and IL-1β signalings.

## Data Availability

The authors confirm that the data supporting the findings of this study are available within the article.

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
