# Peer review of "Extracellular Vesicle-Associated MicroRNA-138-5p Regulates Embryo Implantation and Early Pregnancy by Adjusting GPR124"

_pharmaceutics, 2022, doi:10.3390/pharmaceutics14061172_

Round 1

Reviewer 1 Report

The manuscript reported extracellular vesicle-associated microRNA-138-5p plays a role in the embryo implantation and early pregnancy by modulating GPR124 and inflammasome, NLRP3. A variety of analysis results is included to support their aims. However, there are some issues that need to be revised before the publication.

  1. Duplicate descriptions should be revised properly in the whole manuscript.
  • For example, (Abstract) “Transmission electron microscopy (TEM) images of iso-lated extracellular vesicles were obtained.” and “Isolated extracellular vesicles from decidua and decidual stro-mal cells were confirmed by TEM images.
  1. There are some descriptions that are not appropriate. For example, “Transmission electron microscopy (TEM) images of isolated ex-tracellular vesicles were performed.”, etc.
  2. Abbreviations should be carefully used and should be appropriately modified. The abbreviation must appear the first time when it is used. Inappropriate use of abbreviations hinders the readability.
  3. There are no markings A, B, and C in Figure 1.
  4. What is the concentration of particles (extracellular vesicles) through NTA analysis?
  5. In the result section and figure caption, the description of methods is redundant and should be revised.
  6. In Figure 4A, scale bar is missing. There was a difference in the 2nd Ab only staining of the spontaneous groups (row 2 and 3). What does this mean?

Author Response

Dear reviewers and editors :

Thanks for your valuable correction. The responses to the comments are described below. Thanks a lot!

Sincerely,

Authors

 Responses to the commemts:

Reviewer 1

  1. Duplicate descriptions should be revised properly in the whole manuscript.
  • For example, (Abstract) “Transmission electron microscopy (TEM) images of iso-lated extracellular vesicles were obtained.” and “Isolated extracellular vesicles from decidua and decidual stro-mal cells were confirmed by TEM images.”

 Ans : We revised the description as suggestion.

  1. There are some descriptions that are not appropriate. For example, “Transmission electron microscopy (TEM) images of isolated ex-tracellular vesicles were performed.”, etc.

 Ans : We revised the description as suggestion.

  1. Abbreviations should be carefully used and should be appropriately modified. The abbreviation must appear the first time when it is used. Inappropriate use of abbreviations hinders the readability.

 Ans : We revised the description as suggestion.

  1. There are no markings A, B, and C in Figure 1.

 Ans : We addressed the comments as suggestion.

  1. What is the concentration of particles (extracellular vesicles) through NTA analysis?

 Ans : We addressed the comments as suggestion in M&M 2.4.

  1. In the result section and figure caption, the description of methods is redundant and should be revised.

 Ans : We revised the description as suggestion.

  1. In Figure 4A, scale bar is missing. There was a difference in the 2nd Ab only staining of the spontaneous groups (row 2 and 3). What does this mean?

Ans : We revised the figure 4A and addressed the comments as suggestion.

Reviewer 2 Report

The article "Extracellular vesicle-associated microRNA-138-5p regulates embryo implantation and early pregnancy by adjusting GPR124" is good one.

Major Points.

Please provide all raw images for westren blots for all figure in supplemental files.

The drawn conclusions does not supported by results.

There is no mechanism.

Please change the statements of miR-138-5p and GPR124-regulated NLRP3 inflammasome because it has an association role, not direct role.

Text in figure is not very clear

Quality of figures are not good

Has the author submitted the bioinformatics data to NCBI etc, if yes please provide GEO number.

How the author selected the miR-138-5pd to study why not others??

Author Response

Dear reviewers and editors :

Thanks for your valuable correction. The responses to the comments are described below. Thanks a lot!

Sincerely,

Authors

Responses to the commemts:

Reviewer 2

  1. Please provide all raw images for westren blots for all figure in supplemental files.

Ans :We provide the raw images of western blots in supplemental files as suggestion.

  1. The drawn conclusions does not supported by results.

Ans : We revised the description as suggestion.

  1. There is no mechanism.

Ans : We revised the description as suggestion.

  1. Please change the statements of miR-138-5p and GPR124-regulated NLRP3 inflammasome because it has an association role, not direct role.

Ans : We revised the description as suggestion.

  1. Text in figure is not very clear

Ans : We revised the text in figures as suggestion.

  1. Quality of figures are not good

Ans : We revised the quality of figures as suggestion.

  1. Has the author submitted the bioinformatics data to NCBI etc, if yes please provide GEO number.

Ans : We addressed the comments as suggestion. The GEO number is GSE203420.

  1. How the author selected the miR-138-5pd to study why not others??

Ans : Based on the data showed in figure 2, microarray analysis of microRNAs and RT-qPCR analysis of the expressed miRNA showed that miRNA-138-5p is down-regulated in spontaneous miscarriage compared to normal pregnancy. Therefore, we selected the miR-138-5p to study.

Reviewer 3 Report

In the manuscript “Extracellular vesicle-associated microRNA-138-5p regulates embryo implantation and early pregnancy by adjusting GPR124”, Hsien-Ming Wu et. al. demonstrated the role of extracellular vesicles, miR-138-5p, GPR124, and NLRP3 inflammasome in normal early pregnancy and spontaneous miscarriage. The work can be considered for publication provided following revisions can be carried out.

1) Why the authors chose only microRNA-138-5P. Authors should provide logic behind this in the introduction part.

2) There are many genes controlled be by microRNA-138-5P. Authors should explain why GPR124 has been chose as a target gene.

3) It will be worth to investigate other target genes for microRNA-138-5P.

4) In general, all figures’ quality should be enhanced. The presentation of data is not of standard quality.

5) Language editing is required. There are many typos.

6) The quality of TEM is not good. Authors are also recommended to show the immunogold-EM for CD63, CD9, and any other exosome marker.

7)  In the discussion, the authors are recommended to include a few latest literatures about the exosome-based theranostic techniques, such as LSPR, AFM, and microfluidic drug loading, and write about potential of their work in the context of pharmaceutics application, especially microfluidic drug loading in exosome, liquid biopsy application of exosomes for diagnostic purposes.

Author Response

Dear reviewers and editors :

Thanks for your valuable correction. The responses to the comments are described below. Thanks a lot!

Sincerely,

Authors

 Responses to the commemts:

Reviewer 3

1) Why the authors chose only microRNA-138-5P. Authors should provide logic behind this in the introduction part.

Ans : Based on the data showed in figure 2, microarray analysis of microRNAs and RT-qPCR analysis of the expressed miRNA showed that miRNA-138-5p is down-regulated in spontaneous miscarriage compared to normal pregnancy. Therefore, we selected the miR-138-5p to study. We provide logic behind this in the introduction part as suggestion.

2) There are many genes controlled be by microRNA-138-5P. Authors should explain why GPR124 has been chose as a target gene.

Ans : Target scan, a miRNA target prediction program, is applied to search for putative target genes of miR-138-5p. By using Target scan, GPR124 was predicted as a potential target of miR-138-5p in Figure. 3A. We focused on GPR124 because of its established relevance in angiogenesis and atherosclerosis possibly related to embryo implantation. We provide the description as suggestion in results 3.3.

3) It will be worth to investigate other target genes for microRNA-138-5P.

Ans : Yes, we will investigate other target genes in future ongoing projects.

4) In general, all figures’ quality should be enhanced. The presentation of data is not of standard quality.

Ans : We revised the quality of figures as suggestion.

5) Language editing is required. There are many typos.

Ans : We have our manuscript checked by a native English-speaking colleague.

6) The quality of TEM is not good. Authors are also recommended to show the immunogold-EM for CD63, CD9, and any other exosome marker.

Ans : We revised the figure 1 and addressed the comments as suggestion.

7)  In the discussion, the authors are recommended to include a few latest literatures about the exosome-based theranostic techniques, such as LSPR, AFM, and microfluidic drug loading, and write about potential of their work in the context of pharmaceutics application, especially microfluidic drug loading in exosome, liquid biopsy application of exosomes for diagnostic purposes.

Ans : We addressed the comments as suggestion in the discussion.

Round 2

Reviewer 1 Report

No more comments

Reviewer 2 Report

no more comments

Reviewer 3 Report

Satisfactory revision.